# Utilization of Ureteral Access Sheath in Retrograde Intrarenal Surgery: A Systematic Review and Meta-Analysis

**DOI:** 10.3390/medicina60071084

**Published:** 2024-07-02

**Authors:** Chi-Bo Lin, Shu-Han Chuang, Hung-Jen Shih, Yueh Pan

**Affiliations:** 1Divisions of Urology, Department of Surgery, Changhua Christian Hospital, Changhua 500, Taiwan; 182985@cch.org.tw; 2Division of General Practice, Department of Medical Education, Changhua Christian Hospital, Changhua 500, Taiwan; b101106100@tmu.edu.tw; 3Department of Post-Baccalaureate Medicine, College of Medicine, National Chung Hsing University, Taichung 402, Taiwan; 4Department of Urology, School of Medicine, College of Medicine, Taipei Medical University, Taipei 110, Taiwan; 5Ph.D. Program in Translational Medicine, National Chung Hsing University, Taichung 402, Taiwan; 6Rong Hsing Research Center for Translational Medicine, National Chung Hsing University, Taichung 402, Taiwan

**Keywords:** ureteral access sheath, retrograde intrarenal surgery, stone-free rate, meta-analysis

## Abstract

*Background and Objectives*: This paper evaluates the efficacy and safety of ureteral access sheath (UAS) utilization in retrograde intrarenal surgery (RIRS). *Materials and Methods*: We searched PubMed, Embase, and the Cochrane Library up to 30 August 2023. The inclusion criteria comprised English-language original studies on RIRS with or without UAS in humans. The primary outcome was SFR, while the secondary outcomes included intraoperative and postoperative complications, the lengths of the operation and the hospitalization period, and the duration of the fluoroscopy. Subgroup analyses and a sensitivity analysis were performed. Publication bias was assessed using funnel plots and Egger’s regression tests. Dichotomous variables were analyzed using odds ratios (ORs) with 95% confidence intervals (CIs), while mean differences (MDs) were employed for continuous variables. *Results*: We included 22 studies in our analysis. These spanned 2001 to 2023, involving 12,993 patients and 13,293 procedures. No significant difference in SFR was observed between the UAS and non-UAS groups (OR = 0.90, 95% CI 0.63–1.30, *p* = 0.59). Intraoperative (OR = 1.13, 95% CI 0.75–1.69, *p* = 0.5) and postoperative complications (OR = 1.29, 95% CI 0.89–1.87, *p* = 0.18) did not significantly differ between the groups. UAS usage increased operation times (MD = 8.30, 95% CI 2.51–14.10, *p* = 0.005) and fluoroscopy times (MD = 5.73, 95% CI 4.55–6.90, *p* < 0.001). No publication bias was detected for any outcome. *Conclusions*: In RIRS, UAS usage did not significantly affect SFR, complications, or hospitalization time. However, it increased operation time and fluoroscopy time. Routine UAS usage is not supported, and decisions should be patient-specific. Further studies with larger sample sizes and standardized assessments are needed to refine UAS utilization in RIRS.

## 1. Introduction

Urinary stone disease remains a significant focus in urology practice worldwide, with prevalence ranging from 1% to 15% [1]. The primary goal of minimally invasive treatments for kidney stones is to achieve stone-free status, with minimal tissue damage. With advancements such as flexible ureterorenoscopes and laser lithotriptors, retrograde intrarenal surgery (RIRS) has emerged as a crucial alternative treatment for stone management [2]. Currently, both RIRS and extracorporeal shockwave lithotripsy are recommended as the initial approaches for treating kidney stones smaller than 2 cm according to EAU Guidelines [3].

The ureteral access sheath (UAS), initially introduced by Hisao Takayasu in 1974 [4], represents a useful modality for urologists, offering facilitated retrograde access to the upper urinary tract during endoscopic procedures, particularly for RIRS. These devices, designed to ease the passage of endoscopes and allow repeated entry and exit from the ureter, have gained popularity over the years due to their perceived advantages in improving visibility, reducing intrarenal pressure during irrigation, and potentially enhancing stone fragmentation and retrieval [5]. Meanwhile, the literature has raised concerns about increasing costs, the potential for ureteral injury, and the risk of developing long-term ureteral strictures for UAS usage [5]. While UAS utilization has become increasingly common in urological practice, the effectiveness and complications related to UAS placement remain a subject of considerable debate [6].

With developments in technology, UAS devices have been produced with various characteristics, including different lengths, calibers, hydrophilicity characteristics, materials, and stiffnesses. However, the routine utilization of UASs during RIRS has been a topic of debate and controversy. Under the recommendations from the European Association of Urology Guidelines, it is not clear whether a UAS should be used during ureteroscopy [7]. However, the American Urological Association’s guidelines recommend the utilization of UASs for RIRS in cases involving complex and high-volume kidney stones [8].

In a recent meta-analysis in 2018, the use of UASs compared to the non-use of UASs indicated no disparities in stone-free rates, operative times, hospitalization durations, or intraoperative complications. However, it did significantly elevate the occurrence of postoperative complications [9], indicating no apparent advantages to using a UAS during ureteroscopy. However, numerous studies and randomized controlled trials (RCTs) have been published recently that have indicated the necessity of an update [10,11,12,13,14,15,16,17,18,19,20,21,22,23]. Due to the lack of clear clinical evidence, the development of guidelines, physician training, and communication with patients will all become challenging. By enroling the latest studies, we aimed to update the current evidence for clinical practice and clarify the benefit of the usage of UASs.

## 2. Materials and Methods

### 2.1. Study Design

Preferred Reporting Items for Systematic Reviews and Meta-Analyses (PRISMA) guidelines were adhered to during the execution of this meta-analysis [24,25]. Our present study was registered and is available on PROSPERO (CRD42022360852). During the entire research process, encompassing a literature search, data extraction, quality assessments, and a statistical analysis, two authors collaborated independently and cross-verified their efforts. In cases of disagreements, a third reviewer was consulted for resolution.

### 2.2. Search Strategy

Two authors independently searched (up to 30 August 2023) PubMed, Embase, and the Cochrane Library using the terms “ureteroscopy” and “ureteral access sheath” with the Medical Subject Heading (MeSH) search strategies. In accordance with the Harvard Countway Library (https://guides.library.harvard.edu/meta-analysis; accessed on 30 August 2023), methodology filters were derived for the study design. Potentially eligible articles were manually screened from the reference lists of included articles. The retrieval strategy did not restrict retrieval by publication date.

### 2.3. Eligibility Criteria

Duplicate articles were removed after the completion of the searches. The criteria for study inclusion were as follows: (i) studies reporting the outcomes of using UASs when performing RIRS in patients with renal stones; (ii) studies presented in English; (iii) journal articles that were original; (iv) comparative studies about surgeries with or without UASs; and (v) in vivo human studies. A further exclusion was made for articles that did not clearly define the procedure or did not provide adequate data.

### 2.4. Data Extraction and Outcome Measurements

A well-crafted and uniformly structured form was used to retrieve the relevant data from either the datasets or the graphs provided in the reviewed studies. For each article, data extraction encompassed various elements, including the last name of the first author, publication year, sample sizes, types of procedures performed, study period, country of origin, follow-up duration, article type, patient age, stone burden, procedures involving multiple stones, preoperative or postoperative stent placement, type of ureteroscopy, the definition of stone-free rate (SFR), determination of UAS usage, and outcomes. The primary outcome of interest was SFR, as this metric directly reflects the success of stone removal and patient outcomes. Although UAS offers several benefits, such as reduced intrarenal pressure, improved visibility, and prolonged device lifespan, the SFR provides a clear, quantifiable measure of procedural efficacy. Our focus on SFR allowed us to provide robust data on the effectiveness of RIRS with UASs, thereby offering valuable insights for clinical practice. The emphasis on SFR does not diminish the importance of other UAS benefits, which are also discussed as secondary outcomes, including intraoperative and postoperative complications, the lengths of the operation and the hospitalization period, and the duration of the fluoroscopy.

### 2.5. Quality Assessments

We evaluated and assigned methodological quality scores to each study using the Jadad scale for RCTs and the Newcastle Ottawa Scale (NOS) for non-RCTs [26,27].

### 2.6. Statistical Analysis

The data extracted for the analysis were processed using Comprehensive Meta-Analysis software, version 3.3, to conduct the meta-analysis. With 95% confidence intervals (CIs), the SFRs and intraoperative and postoperative complications were calculated as dichotomous variables using odds ratios (ORs), while the operative time, hospitalization time, and fluoroscopy time were calculated as continuous variables using mean differences (MDs). The analysis of pooled effects was conducted using the Z-test. A *p*-value of below 0.05 was regarded as indicative of statistical significance. The total numbers of patients were multiplied by the complication rates in studies that did not provide the exact number of patients with complications.

To assess statistical heterogeneity among the studies, the chi-squared test (Chi^2^), Cochrane Q test, and I-squared statistic test (I^2^) were used. The presence of heterogeneity is indicated when a *p*-value is less than 0.1 in the Cochrane Q test or when I^2^ values are > 50%. Instead of employing a fixed-effect model, a random-effects model was utilized, assuming heterogeneity across all the gathered studies [28].

For the primary outcome, subgroup analyses were performed following the study designs. By removing individual studies one at a time, a sensitivity analysis was conducted to evaluate the impacts of potential outliers. To explore publication bias, funnel plots were created and Egger’s regression tests were conducted when there were at least three rows of data [29]. If publication bias was indicated, a trim and fill analysis was further performed [30].

## 3. Results

### 3.1. Study Selection

After searching across three databases (PubMed, Embase, and the Cochrane Library), a total of 700 studies were initially identified. Following the removal of 232 duplicates, we reviewed the titles and abstracts, resulting in 26 articles that underwent a full-text assessment for eligibility. Among these, one was excluded for an unclear definition of its procedures and two were excluded for unavailable data, while one consisted of a duplicated cohort from another, and one with a larger sample size and intact data was further included. Finally, twenty-two eligible studies were included in this meta-analysis, as shown Figure 1 [10,11,12,13,14,15,16,17,18,19,20,21,22,23,31,32,33,34,35,36,37,38].

### 3.2. Study Characteristics

Published from 2001 to 2023, the selected studies enrolled 12,993 patients, with a total of 13,293 procedures, synthesizing a UAS group of 7328 procedures and a non-UAS group of 5965 procedures (Table 1). Among these studies, eighteen reported SFRs, eight reported intraoperative complications, thirteen reported postoperative complications, twelve reported operative times, five reported hospitalization times, and two reported fluoroscopy times. There were six RCTs [20,21,22,23,35,38], five prospective cohort studies [11,14,16,31,32], and eleven retrospective cohort studies [10,12,13,15,17,18,19,33,34,36,37] classified from the included studies. The quality scores of the non-randomized studies ranged from 5 to 6, while the RCTs were scored between 1 and 4. The characteristics of the included studies are summarized in Appendix A.

### 3.3. Primary Outcomes

#### 3.3.1. SFR

Fourteen studies stated the definition of SFR [11,13,14,15,16,20,21,22,23,31,32,33,35,36]. Three described it as completely clean [14,35,36], while one [32], three [13,16,31], and seven [11,15,20,21,22,23,33] identified SFRs when having residual stones of less than 1, 2, and 3 mm, respectively. Radiographs were performed to ascertain the stone-free statuses. Of the twenty-two included studies, eighteen comprising 8237 RIRS procedures compared SFRs between UAS and non-UAS groups. Our findings revealed no significant differences between the groups (OR = 0.90, 95% CI 0.63–1.30, *p* = 0.59, Figure 2). However, significant heterogeneity was observed (Q = 86.14, *p* < 0.001, I^2^ = 80.26%). Our sensitivity analysis confirmed that the individual studies did not notably impact our overall findings, affirming the reliability of our analysis.

#### 3.3.2. Subgroup Analysis

As per the study design, subgroup analyses were conducted to further assess the effectiveness of UASs in SFRs (Table 2). The results were similar to the original outcome, indicating the robustness and validity of our findings. Furthermore, the insignificant heterogeneity of the subgroup of RCTs was demonstrated.

### 3.4. Secondary Outcomes

#### 3.4.1. Intraoperative Complications

Eight studies provided data on intraoperative complications [10,14,19,21,22,33,34,35] comprising bleeding, perforation of the ureteral or calyceal systems, the Post-Ureteroscopic Lesion Scale (PULS) classification system [39], failed ureteroscopy, etc. The results of these 8355 procedures revealed no significant differences between the groups (OR = 1.13, 95% CI 0.75–1.69, *p* = 0.5, Figure 3), with moderate heterogeneity (Q = 14.65, *p* = 0.04, I^2^ = 52.21%).

#### 3.4.2. Postoperative Complications

Thirteen studies reported postoperative complications in two groups [10,11,13,15,16,18,20,21,22,31,32,33,38], including 5697 procedures. The identifications were made using systemic inflammatory response syndrome, gross hematuria, aspiration pneumonia, Clavien classification [40], fever, positive urinary culture, positive blood culture, urosepsis, urinary tract infection, etc. The findings showed no significant differences (OR = 1.29, 95% CI 0.89–1.87, *p* = 0.18, Figure 4) and moderate heterogeneity (Q = 22.78, *p* = 0.03, I^2^ = 47.31%).

#### 3.4.3. Operation Time

Our findings in twelve studies, including 6676 procedures, revealed that using a UAS significantly increased the operation time (MD = 8.30, 95% CI 2.51–14.10, *p* = 0.005, Figure 5), and there was significant heterogeneity (Q = 157.53, *p* < 0.001, I^2^ = 93.02%) [11,12,13,15,16,17,18,20,21,32,33,35].

#### 3.4.4. Hospitalization Time

In five articles [15,16,21,32,33], 4705 procedures were examined to assess the hospitalization times. Pooling the data from these articles, the results showed insignificant differences between the groups that had operations with or without UASs (MD = −0.03, 95% CI −0.17–0.11, *p* = 0.68, Figure 6), with low heterogeneity (Q = 5.75, *p* = 0.22, I^2^ = 30.48%).

#### 3.4.5. Fluoroscopy Time

Only two articles reported the differences in the fluoroscopy times [13,21], consisting of 173 procedures. The results indicated a significant difference between the groups (MD = 5.73, 95% CI 4.55–6.90, *p* < 0.001, Figure 7) and no heterogeneity (Q = 0.58, *p* = 0.45, I^2^ = 0.00%). This led us to conclude that the UAS group received significantly longer fluoroscopy times than the non-UAS group.

### 3.5. Publication Bias

We assessed publication bias when a forest plot included three or more studies. Visualizations of the funnel plots (Appendix A) and Egger’s tests were conducted, indicating no evidence of publication bias in the SFRs (*p* = 0.48), intraoperative (*p*  = 0.62) and postoperative (*p* = 0.38) complications, operation times (*p* = 0.87), and hospitalization times (*p* = 0.69).

## 4. Discussion

The use of a UAS in an RIRS has significant clinical implications that warrant thorough discussion. A UAS facilitates the repeated passage of instruments into the kidney, thereby reducing intrarenal pressure and improving visibility and irrigation. By providing these benefits, a UAS was considered to play a critical role in the success and safety of RIRS procedures. Our study highlights the evidence from UAS usage, aiming to underscore its necessity in clinical practice and encourage further research on its practical applications. Our findings seek to provide a foundation for future studies to build upon, ensuring that the benefits of UAS usage are fully recognized and utilized in urological surgeries.

From the pooled results of 22 studies, we found no statistical differences in SFRs, intraoperative and postoperative complications, and hospitalization times between the UAS and non-UAS groups. However, significantly longer operation times (MD = 8.30, 95% CI 2.51–14.10, *p* = 0.005) and fluoroscopy times (MD = 5.73, 95% CI 4.55–6.90, *p* < 0.001) were identified in the UAS group compared to the non-UAS group. A subgroup analysis was performed for further investigation, showing insignificant differences in the study designs and revealing the stability of this meta-analysis. Low heterogeneity of the subgroup of RCTs was found (Q = 5.75, *p* = 0.33, I^2^ = 13.06%), showing the robustness of the evidence even when the decision about UAS usage was based on randomization rather than surgeon preference. Moreover, no publication bias was detected after applying an Egger’s test. The results of the present study did not demonstrate an obvious benefit to using a UAS during ureteroscopy. This indicates that its use should not be a routine practice in all cases.

Five years have passed since the publication of the meta-analysis by Huang et al. [9]. In contrast, our investigation incorporated data from 14 more studies. While the results regarding most outcomes remained consistent, the updated findings indicated that the previously observed higher risk of postoperative complications in the UAS group is no longer statistically significant. This finding was consistent with the review by Asutay et al. [41], concluding that the use of a ureteral access sheath does not increase ureteral injury. We hypothesized that this change may be attributed to factors such as advancements in UAS materials, technology, and sizes of options. Additionally, the maturation of the techniques and procedures themselves may have played a role in this observed trend.

Regarding the decision-making process for the utilization of a UAS, some studies employed random allocation, whereas others relied on the preferences and judgments of operators. The physicians based their decisions on various factors such as the degree of ureteral stenosis, stone size, the intention to maximize stone-fragment retrieval, and the anticipated number of instrument insertions, potentially introducing bias into the grouping process. Within our subgroup analysis, we observed that the pooled results from the six RCTs not only exhibited no differences between the two groups but also displayed a substantial reduction in heterogeneity. This decrease in heterogeneity might be attributed to the randomized allocation, which significantly lowered the risk of bias stemming from operator preferences.

Several hypotheses have been proposed to support the routine use of UASs. First, it has been suggested that UASs could enhance SFRs by allowing urologists to enter and exit the urinary collecting system rapidly [36]. Second, it has been theorized that a UAS could reduce intrarenal pressure by facilitating the flow of fluids out of the collecting system [42,43], potentially lowering the risk of clinical complications. However, in our study, neither SFR nor the risk of complications exhibited statistically significant differences. In other words, these purported benefits did not manifest in our meta-analysis. It is possible that variations in the UASs themselves, study designs, and patient populations may have influenced such outcomes, but we did not identify clinical evidence to advocate for the routine use of UASs.

On the other hand, the literature has raised concerns about radiation exposure during an RIRS. Jamal et al. reported the substantial radiation exposure faced by kidney stone patients, particularly through multiple CT scans and fluoroscopies in the periprocedural phase [44]. Additionally, Park et al. established noteworthy associations between radiation exposure, stone number, and Hounsfield unit value [45]. In our investigation, we observed considerable lengthening of the surgical procedures when employing UASs and a notable increase in fluoroscopy exposure. Our findings suggest that routine UAS usage may raise the risk of increased radiation exposure for both patients and urologists. This extension in procedure duration likely stems from the precise positioning and adjustment of a UAS which, while recognized for improving access and visualization, also prolongs the time spent under fluoroscopy.

There are several limitations associated with this meta-analysis study. First, there was variability across the articles concerning stone size, location, and quantity, which diminished the precision of the results and potentially contributed to increased heterogeneity. Second, a limited number of articles was available for certain outcomes, such as hospitalization duration or fluoroscopy time. We advocate for future research to incorporate relevant statistical analyses and investigations to enhance our understanding of the pros and cons of UASs. Third, it is possible that overall longer fluoroscopies and total operative times were primarily associated with the insertion and removal of the UASs. However, considering the comprehensive cost considerations regarding time and operating room utilization, we have faithfully presented these data. Lastly, diverse instruments were employed for the assessments of complications in the different studies, potentially resulting in high heterogeneity in the evaluation process. In our study, we aimed to examine the effects of UAS usage on cost, patient comfort, shared decision-making processes, and outcome across the different subgroups. However, the existing data do not support such detailed analyses. Our work highlights this gap, and we urge future researchers to prioritize evaluating UAS usage and conduct more practical analyses in this area.

Despite this, there is currently insufficient evidence to support the routine usage of UASs. Clinicians should exercise caution in individual cases when considering the use of UASs. Moreover, they should engage in thorough communication and explanations and shared decision-making with patients. Further research on the timing and patient selection for UASs is warranted, with larger sample sizes, more standardized assessments, and well-designed adjustments for confounders.

## 5. Conclusions

Our results demonstrated that using UASs in RIRSs did not affect SFRs, intraoperative and postoperative complications, and hospitalization times. Instead, significantly longer operation times and fluoroscopy times were identified in the UAS group compared to the non-UAS group, despite the limited number of available data.

## Figures and Tables

**Figure 1 medicina-60-01084-f001:**
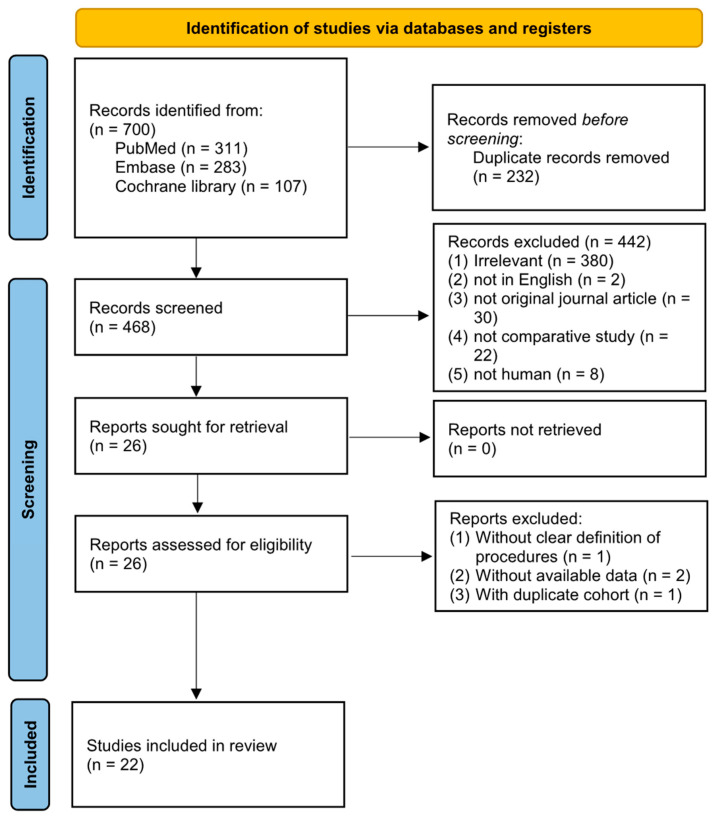
Flowchart of the study selection.

**Figure 2 medicina-60-01084-f002:**
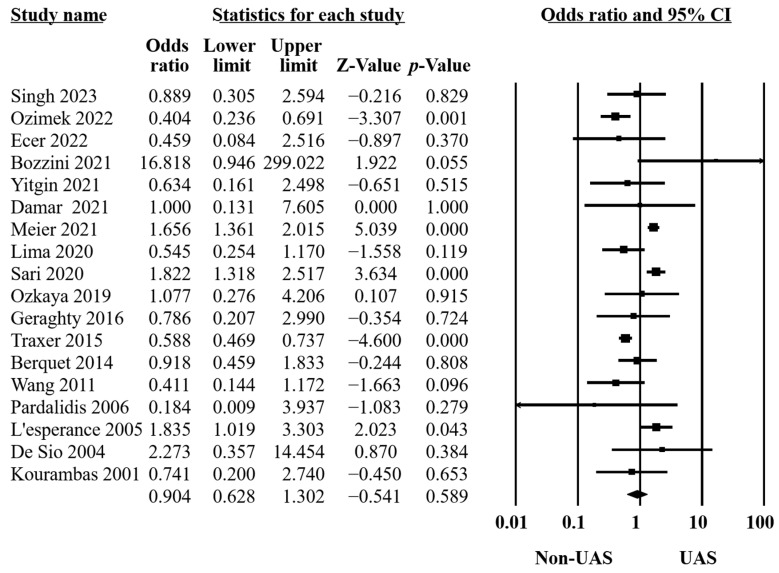
Forest plot of the SFRs [10,11,13,14,15,16,20,21,22,23,31,32,33,34,35,36,37,38].

**Figure 3 medicina-60-01084-f003:**
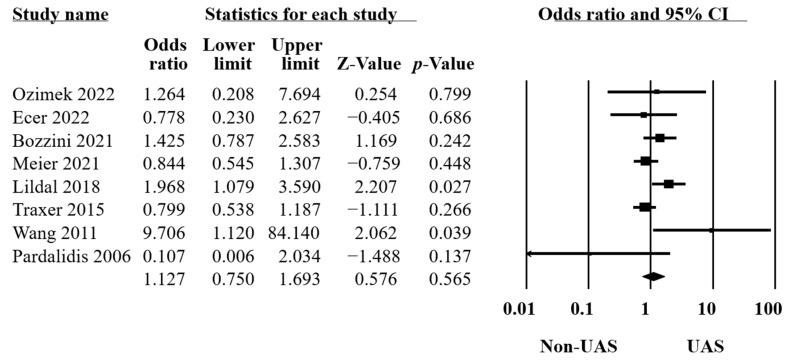
Forest plot of the intraoperative complications [10,14,19,21,22,32,34,35].

**Figure 4 medicina-60-01084-f004:**
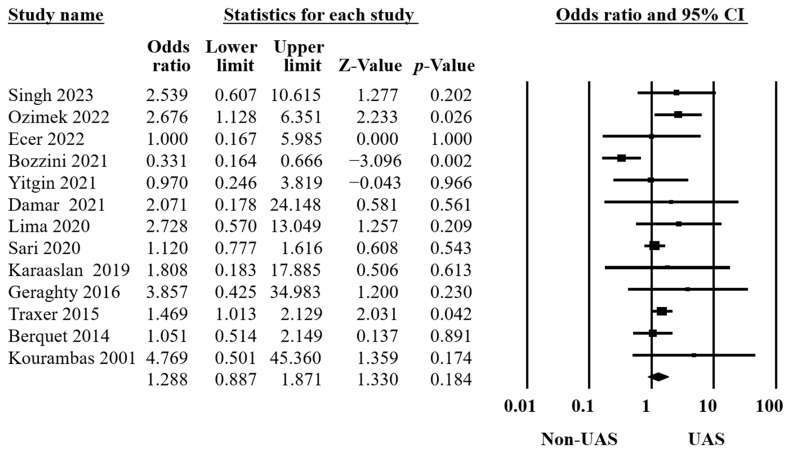
Forest plot of the postoperative complications [10,11,13,15,16,18,20,21,22,31,32,33,38].

**Figure 5 medicina-60-01084-f005:**
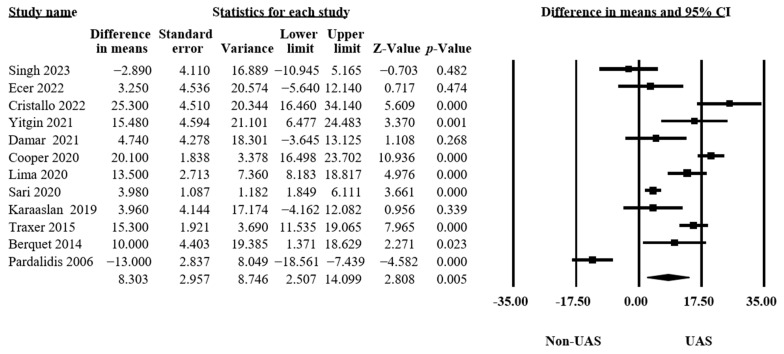
Forest plot of the operation times [11,12,13,15,16,17,18,20,21,32,33,35].

**Figure 6 medicina-60-01084-f006:**
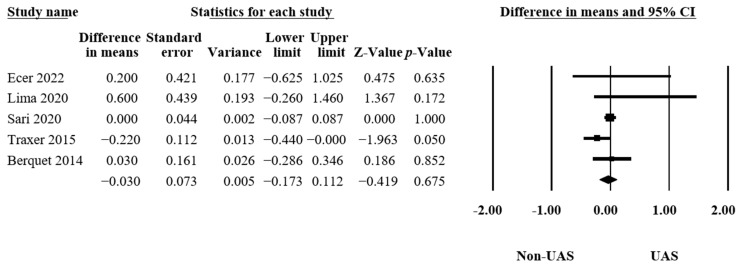
Forest plot of the hospitalization times [15,16,21,32,33].

**Figure 7 medicina-60-01084-f007:**
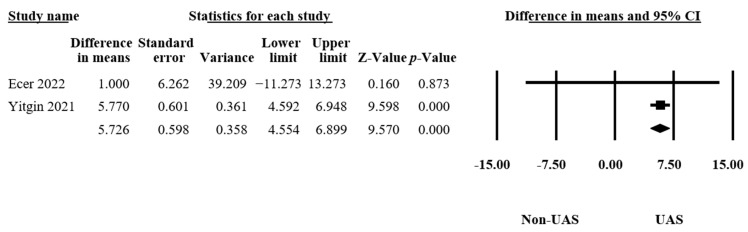
Forest plot of the fluoroscopy times [13,21].

**Table 1 medicina-60-01084-t001:** Summary of the included studies.

Author	Year	Sample Sizes/Procedures, n	Study Period	Country	Follow-Up Time	Article Type	Quality Score
Singh [20]	2023	81	July 2019–December 2021	India	1 month	RCT	4 ^b^
Ozimek [10]	2022	283	September 2013–June 2017	Germany	14 days	Retrospective cohort	6 ^a^
Ecer [21]	2022	60	NR	Turkey	14 days	RCT	1 ^b^
Cristallo [12]	2022	241	January 2018–May 2020	Argentina	3 months	Retrospective cohort	6 ^a^
Bozzini [22]	2021	181	January 2017–December 2017	Italy and Spain	3 days	RCT	3 ^b^
Yitgin [13]	2021	113	February 2019–May 2020	Turkey	3 months	Retrospective cohort	6 ^a^
Damar [11]	2021	60	February 2017–November 2017	Turkey	1 month	Prospective cohort	6 ^a^
Meier [14]	2021	5229	June 2016–July 2018	USA	60 days	Prospective cohort	6 ^a^
Cooper [17]	2020	1060/1332	January 2012–September 2016	USA	8 weeks	Retrospective cohort	6 ^a^
Lima [16]	2020	338	March 2012–July 2018	UK	2–3 months	Prospective cohort	6 ^a^
Sari [15]	2020	1808	2012–2019	Turkey	3 months	Retrospective cohort	6 ^a^
Karaaslan [18]	2019	129	January 2016–October 2018	Turkey	NR	Retrospective cohort	5 ^a^
Özkaya [23]	2019	131	January 2017–June 2018	Turkey	1 month	RCT	2 ^b^
Lildal [19]	2018	180	November 2013–February 2016	Denmark	At the end of each procedure	Retrospective cohort	5 ^a^
Geraghty [31]	2016	43/68	March 2012–October 2014	UK	2–3 months	Prospective cohort	6 ^a^
Traxer [32]	2015	2239	January 2010–October 2012	France	NR	Prospective cohort	5 ^a^
Berquet [33]	2014	280	2009–2012	France	1–3 months	Retrospective cohort	7 ^a^
Wang [34]	2011	96	1999–2009	USA	11 months	Retrospective cohort	5 ^a^
Pardalidis [35]	2006	98	January 2001–December 2004	Greece	1 year	RCT	2 ^b^
L’esperance [36]	2005	256	1997–2003	USA	2 months	Retrospective cohort	7 ^a^
De Sio [37]	2004	28	1999–May 2003	Switzerland	NR	Retrospective cohort	5 ^a^
Kourambas [38]	2001	59/62	October 1999–January 2000	USA	3 months	RCT	2 ^b^

^a^ Assessment of the quality of the cohort studies using the Newcastle-Ottawa Scale (score from 0 to 9). ^b^ Assessment of the quality of the RCTs using the Jadad scale (score from 0 to 5). RCT, randomized controlled trial; NR, not reported.

**Table 2 medicina-60-01084-t002:** Summary of included studies.

Subgroup	Studies, n	Heterogeneity			Odds Ratios (95% Confidence Interval)	*p*-Value
		Cochrane Q	*p*-Value	I-Squared, %		
Original	18	86.14	<0.001	80.26	0.90 (0.63, 1.30)	0.59
Study design						
RCT	6	5.75	0.33	13.06	0.88 (0.44, 1.75)	0.5
Retrospective cohort	7	30.05	<0.001	80.03	0.95 (0.52, 1.73)	0.86
Prospective cohort	5	48.91	<0.001	91.82	0.84 (0.42, 1.71)	0.64

RCT, randomized controlled trial.

## Data Availability

The original contributions presented in the study are included in the article/Appendix A, and further inquiries can be directed to the corresponding author(s).

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
