# Peer review of "Utilization of Ureteral Access Sheath in Retrograde Intrarenal Surgery: A Systematic Review and Meta-Analysis"

_medicina, 2024, doi:10.3390/medicina60071084_

Round 1

Reviewer 1 Report

Comments and Suggestions for Authors

The authors addressed the efficacy and safety of ureteral access sheath (UAS) utilization in retrograde intrarenal surgery (RIRS). They relied on a meta-analysis methodology including 22 studies (either RCT or prospective/retrospective). Specifically, UAS did not significantly affect the stone-free rate, or the occurrence of either intraoperative or postoperative complications. Conversely, it only prolonged the operation time and the fluorescence time. Thus, no clear guidelines should be resumed since the approach should be tailored to the patient's characteristics. 

Any additional data on the patient's characteristics of thee studies? And on UAS materials? 

Any data available on trial on the role of UAS in the RIRS procedure?

Any data available on costs or specific subset of the population, such as pregnants (PMID= 38049673)

The discussion appears only speculative with no clinical implication. The authors should better define the background of RIRS depicting the importance of using a UAS. If this aspect cannot be fully evaluated, the study itself loses its importance. 

Author Response

Comments 1: Any additional data on the patient's characteristics of thee studies? And on UAS materials?
Response 1:
Thank you for your insightful comments and suggestions. We have provided a comprehensive summary of the patient's characteristics in Table 1 and Supplementary Material Table S1, including age, stone size, single or multiple stones, and the presence of preoperative and postoperative stents. Additionally, the characteristics of UAS are detailed, indicating whether flexible, semirigid, or rigid ureteroscopy was used, and the decision-making process for UAS usage is explained. We have endeavored to include all pertinent data presented in the studies. If there are specific factors you believe need further clarification, please let us know, and we will be happy to address them. Thank you once again for your valuable feedback.

Comments 2: Any data available on trial on the role of UAS in the RIRS procedure?
Response 2:
Thank you for your inquiry regarding the trials on the role of UAS in the RIRS procedure. We have diligently searched all major literature databases, including PubMed, Embase, and Cochrane Library. Our search strategy included relevant RCTs, retrospective cohort studies, and prospective cohort studies that met our inclusion criteria. Any available data from ongoing trials would have been systematically reviewed and included if they met our criteria. Thank you for your valuable feedback.

Comments 3: Any data available on costs or specific subset of the population, such as pregnants (PMID= 38049673)?
Response 3: 
Thank you for your insightful comments. While we aim to analyze the impact of UAS usage on costs, patient comfort, shared decision-making processes, and outcomes in various subgroups such as pregnant women, the current available data does not permit such analyses. One of the objectives of publishing this article is to raise awareness about this issue, encouraging future researchers to focus on the necessity of UAS usage and to conduct more practical analyses. We will include this point in the discussion section. Thank you for your valuable insights.

Comments 4: The discussion appears only speculative with no clinical implication. The authors should better define the background of RIRS depicting the importance of using a UAS. If this aspect cannot be fully evaluated, the study itself loses its importance.
Response 4: 
Thank you for your valuable feedback. We acknowledge the need to provide a clearer definition of the background of RIRS and the importance of using a UAS. We will enhance the discussion to include more concrete clinical implications, emphasizing the benefits and necessity of UAS in RIRS procedures. By doing so, we aim to strengthen the study's relevance and importance. Your insights are greatly appreciated, and we will ensure this aspect is thoroughly addressed in the revised manuscript.

Reviewer 2 Report

Comments and Suggestions for Authors

Thank you for submitting your work to this journal.

My first idea is that you set your primary objective on the SFR, which is not indicated in the title. The UAS has multiple known benefits, and we should not forget the lifespan of the device itself. Please include a comment on why you focused on the SFR in your analysis.

Since your analysis looks at multiple aspects of using the UAS, maybe you even consider adjusting the title a little bit (please don't see this as a requirement but rather as a piece of advice).

I would also speculate that the overall longer fluoroscopy and total operative time are directly linked to the insertion (and maybe extraction) of the UAS. Probably, the operation itself takes the same time with or without the sheath.

There are some typos which need to be addressed.

Otherwise, a good paper.

I would appreciate if you read and cite the following paper: https://journal.iem.pub.ro/rrst-ee/article/view/109

Comments on the Quality of English Language

minor typos

Author Response

Comments 1: My first idea is that you set your primary objective on the SFR, which is not indicated in the title. The UAS has multiple known benefits, and we should not forget the lifespan of the device itself. Please include a comment on why you focused on the SFR in your analysis.
Response 1:
Thank you for your insightful comments. The decision to concentrate on SFR stems from its critical role as a direct measure of procedural success in RIRS. While UAS indeed offers multiple benefits, including extended device lifespan and enhanced procedural efficiency, the SFR provides a quantifiable and clinically relevant endpoint that directly impacts patient outcomes. We will include a detailed explanation in the manuscript to clarify our rationale for emphasizing SFR in our analysis. Your feedback is greatly appreciated.

Comments 2: Since your analysis looks at multiple aspects of using the UAS, maybe you even consider adjusting the title a little bit (please don't see this as a requirement but rather as a piece of advice).
Response 2:
Thank you for your thoughtful suggestion regarding the title adjustment. While our primary objective is to evaluate the SFR, we aim to provide a comprehensive assessment of the various advantages and disadvantages of UAS use in RIRS. By maintaining the current title, we hope to ensure that readers fully appreciate the multifaceted nature of our analysis and can evaluate the overall impact of UAS on multiple clinical outcomes. We believe this holistic approach will offer a more balanced perspective on the utility of UAS in clinical practice. We greatly appreciate your advice and will consider it for future revisions.

Comments 3: I would also speculate that the overall longer fluoroscopy and total operative time are directly linked to the insertion (and maybe extraction) of the UAS. Probably, the operation itself takes the same time with or without the sheath.
Response 3:
Thank you for your observation. Indeed, it is possible that the overall longer fluoroscopy and total operative time could be directly associated with the insertion (and potentially extraction) of the UAS. However, considering the comprehensive cost implications of time and operating room utilization, we have faithfully presented these data. We will acknowledge this point in the discussion section. Thank you.

Comments 4: There are some typos which need to be addressed.
Response 4:
Thank you for bringing this to our attention. We apologize for any typographical errors present in the manuscript and will diligently address and correct them before final submission. Your careful review is greatly appreciated in ensuring the quality and clarity of our work.

Comments 5: I would appreciate if you read and cite the following paper: https://journal.iem.pub.ro/rrst-ee/article/view/109
Response 5:
Thank you, I have incorporated the suggested paper and found it very informative. It's an excellent piece of work and certainly worthy of citation. Much appreciated.

Reviewer 3 Report

Comments and Suggestions for Authors

The medicina-3074413 is an interesting and appealing topic. The authors present a meta-analysis investigating the role of UAS in terms of safety and efficacy during RIRS.

The abstract is consice, including the necessary infromation about the manuscript.

Introduction

This section is complete. It includes helpful information for the comprehension of the manuscript.

The aim of the study is correctly mentioned.

Materials and Methods

The study design is adequately explained.

The data collection, the inclusion and exclusion criteria are analyzed.

The statistical analysis is correctly explained.

Results

The results are presented in an extensive way.

The figures and tables are important for the completion of the authors' work.

Each subsection is analyzed in a very explenatory way.

Discussion

The discussion is well-written and includes updated data.

The amount of studies conducted regarding the outcomes of UAS use proves the importance of the topic.

The authors correctly commented that the ordinary use of UAS might lengthen the radiation exposure.

The limitations of this meta-analysis are analyzed properly in this section.

Conclusion

The conclusion is representative of the analysis and the work conducted by the authors.

Author Response

Comments: 

The medicina-3074413 is an interesting and appealing topic. The authors present a meta-analysis investigating the role of UAS in terms of safety and efficacy during RIRS.
The abstract is consice, including the necessary infromation about the manuscript.
Introduction
This section is complete. It includes helpful information for the comprehension of the manuscript.
The aim of the study is correctly mentioned.
Materials and Methods
The study design is adequately explained.
The data collection, the inclusion and exclusion criteria are analyzed.
The statistical analysis is correctly explained.
Results
The results are presented in an extensive way.
The figures and tables are important for the completion of the authors' work.
Each subsection is analyzed in a very explenatory way.
Discussion
The discussion is well-written and includes updated data.
The amount of studies conducted regarding the outcomes of UAS use proves the importance of the topic.
The authors correctly commented that the ordinary use of UAS might lengthen the radiation exposure.
The limitations of this meta-analysis are analyzed properly in this section.
Conclusion
The conclusion is representative of the analysis and the work conducted by the authors.

Response:
Thank you for your thoughtful and detailed comments on our manuscript. We are pleased that you find the topic of our meta-analysis on the role of UAS in RIRS interesting and appealing. Your feedback on the abstract, introduction, materials and methods, results, discussion, and conclusion sections is greatly appreciated. We aimed to provide a comprehensive and informative analysis, and we are glad to hear that our efforts have been well-received. Should you have any further suggestions or areas for improvement, please do not hesitate to let us know. Your input is invaluable to us.

Round 2

Reviewer 1 Report

Comments and Suggestions for Authors

The authors addressed my comments properly. Now the methodology is acceptable.

Reviewer 2 Report

Comments and Suggestions for Authors

Thank you for making the changes I suggested.